Subject Areas:
artificial intelligence/molecular computing/nanotechnology

Keywords:
DNA nanotechnology, molecular assembly, chemical reaction networks, quality-diversity algorithms, MAP-elites, genetic algorithms

Author for correspondence:
N. Aubert-Kato
e-mail: naubertkato@is.ocha.ac.jp

# Automated exploration of DNA-based structure self-assembly networks

L. Cazenille[1], A. Baccouche[2] and N. Aubert-Kato[1]

[1]Department of Information Sciences, Ochanomizu University, Tokyo, Japan
[2]CIBIO, University of Trento, Povo, Italy

 LC, 0000-0002-5893-9761; NA-K, 0000-0002-9100-1855

Finding DNA sequences capable of folding into specific nanostructures is a hard problem, as it involves very large search spaces and complex nonlinear dynamics. Typical methods to solve it aim to reduce the search space by minimizing unwanted interactions through restrictions on the design (e.g. staples in DNA origami or voxel-based designs in DNA Bricks). Here, we present a novel methodology that aims to reduce this search space by identifying the relevant properties of a given assembly system to the emergence of various families of structures (e.g. simple structures, polymers, branched structures). For a given set of DNA strands, our approach automatically finds chemical reaction networks (CRNs) that generate sets of structures exhibiting ranges of specific user-specified properties, such as length and type of structures or their frequency of occurrence. For each set, we enumerate the possible DNA structures that can be generated through domain-level interactions, identify the most prevalent structures, find the best-performing sequence sets to the emergence of target structures, and assess CRNs' robustness to the removal of reaction pathways. Our results suggest a connection between the characteristics of DNA strands and the distribution of generated structure families.

## 1. Introduction

DNA nanotechnology has moved forward by leaps and bounds over the past decade [1,2]. The design of nanostructures went from simple motifs based on a repeated pattern of a few complementary strands [3] to massive three-dimensional crystals [4]. Moreover, computer-assisted design has allowed scientists to create intricate structures, ranging from DNA origami [5] to complex three-dimensional shapes [6–8], by leveraging the ability of DNA strands to bind with high selectivity to complementary sequences.

However, design is not a simple process. DNA nanostructures are created dynamically, forming and breaking hydrogen bonds between nucleotides, eventually folding into a specific shape.

**Figure 1.** CRN corresponding to two reversible reactions: $aa^* \Longleftrightarrow aa^*$ (hairpin) and $aa* + aa^* \Longleftrightarrow aa^* \cdot aa^*$.

For that reason, the structure creation process may be kinetically blocked from the premature folding of parts of the structure [9]. Usual approaches aim to reduce the design space by working at the level of structured motifs, such as staple strands and scaffold strand in DNA origami [5] or voxel-based designs in DNA bricks [7,8]. Such approaches focus on rational designs of specific modules that can be further combined into larger structures [10]. However, such designs require controlling the interactions between hundreds of independent sequences, making sequence design difficult. To work around that limitation, researchers have developed algorithms to work with a wider range of DNA scaffolds [11] or allowing the reuse of staple strands, requiring as little as 10 independent sequences for specific shapes [12].

On the other hand, complex interactions between small assemblies of DNA strands can be exploited to create a family of structures from a very limited ($\ll 10$) set of strands. He *et al.* showed that they could create structures ranging from 75 nm tetrahedron to massive 500 nm 'buckyballs' with only three different types of DNA strands, where the final proportion of each structure depends on the respective initial concentration of each type of strand [13]. The assembly process itself is hierarchical: strands are first combined into a basic motif, which can bind to other similar motifs to create larger structures, in turn combining into the final shapes.

Here, we take inspiration from hierarchical self-assembly to introduce a novel framework to identify the categories of structures that DNA strands from a given library can fold into, and the conditions for their emergence. This approach removes *a priori* constraints on the design space, while still allowing the designer to work with a manageable number of DNA strands. The goal of this project is then to automatically explore the design space of libraries of DNA strands. That exploration yields a diverse set of structures of interest that can serve either as inspiration for further refinements or directly as building blocks for larger structures. In particular, we are interested in sets of strands that can assemble into multiple shapes, like those of He *et al.* [13], rather than into a specific target structure.

Methodologies that explore general chemistry systems by searching through chemical reaction networks (CRNs) at the atomic level already exist in the literature [14–17]. These bottom-up approaches aim to iteratively discover the most prevalent species obtained from a set of source species, by relying on heuristic rules of reaction rate, reaction plausibility, and graph theory analysis. However, those methods are too fine-grained to explore self-assembly systems: working at the atom level would translate into very high dimensional search spaces that would be too computationally expensive to explore adequately. As such, we will take inspiration from these methods, but adapt them to work at domain—and sequence—levels of complexity rather than at the level of individual atoms.

We model a self-assembly process by a CRN where each node is a DNA-based structure and connections correspond to binding or separation events between those structures. An example of CRN is given in figure 1. In the rest of this article, we use the term CRN to refer specifically to that type of structure assembly network based on interactions between DNA strands. CRNs encode all reaction pathways (i.e. chains of reactions) between possible structures, thus modelling their assembly and disassembly. We explore the space of CRNs generated by sets of few strands (up to seven). We focus on three libraries of DNA strands, ranging in size from 16 to 1728. The smallest library gives us the possibility to exhaustively evaluate every combination and thus serves as a control for our approach. The second library is a much larger superset of the first. As such, we expect to find at least the structures that are available from the first library. The third library is still larger and inspired by the strands used in the DNA tetrahedron [18].

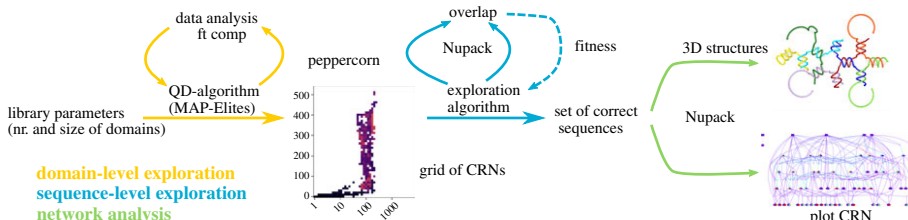

**Figure 2.** General workflow. Our approach is composed of three stages. Domain-level exploration: the MAP-Elites algorithm explores CRNs at the domain-level to find the most promising solutions (according to metrics computed by Peppercorn) and arrange them into a grid of CRNs. Sequence-level exploration: we optimize promising CRNs at the sequence-level to maximize the proportion of well-formed structures (i.e. overlaps between Peppercorn and NUPACK results) they can generate. Network analysis: we plot the resulting CRNs, compute statistical analyses on their graphs and plot examples of the three-dimensional structures they generate.

While optimization methods for DNA nanotechnology have been proposed to generate specific shapes [19], we are interested here in an exploration of the structures that can emerge from CRNs. Methods for the exhaustive domain-level search of the space of CRNs already exist [20], however, they do not scale to larger search spaces and can only be applied for very simple and constrained problems. In this work, only the smallest library can be exhaustively explored. As such, we rely instead on the MAP-Elites algorithm [21], which is capable of exploring large search space efficiently. We combine it with the Peppercorn DNA reaction enumerator [22] to extract the features of each tested CRN. Sets of strands that are expected to produce a wide range of structures are then evaluated with NUPACK [23]. Finally, the CRNs of the most promising candidates are used to generate a three-dimensional model of the predicted structures. Our workflow is shown in figure 2.

We found a variety of behaviours, ranging from simple polymers to complex structures. Finally, we analysed these CRNs using metrics from the network science field (including centrality, clustering and hierarchy measures), to (i) assess the robustness of the CRNs to reaction failure, (ii) find the most influential nodes and pathways and (iii) identify topological properties of the CRNs.

# 2. Domain-level exploration

At this stage, we consider DNA strands at the domain level. A domain is a short sequence of nucleotides that is used as a shorthand in representing strands. Domains are simply represented as a single letter, potentially followed by a* to represent a complementary sequence (kernel notation [22]). Furthermore, independent domains are considered orthogonal, meaning that they do not interact with each other. That simplification is necessary to keep sequences abstract during the enumeration of available reactions (see §2.2).

That enumeration, in turn, yields a CRN, in which we consider DNA structures as vertices and reactions (hybridization, dissociation, three-way branch migration and four-way branch migration) between those as edges. Reaction pathways are the paths between two structures of the CRN.

Finally, a quality-diversity algorithm, MAP-Elites [21], is used to provide a fitness landscape of CRNs across three features: number of independent strands, total number of reactions in the CRN and total number of structures. The exploration is performed over sets of strands selected from a given library.

## 2.1. Libraries and strand sets

We define an $(n, m)$ library as the abstract list of all strands with $n$ domains taken among $m$ possible domain sequences and their complementary sequence. There are thus $2m$ possibilities for each domain on a given strand, which then gives us $n^{2m}$ strands in a $(n, m)$ library.

We investigate the structures that can be created by mixing subsets of specific libraries. By definition, a $(n, m)$ library has $2^{n^{2m}} - 1$ non-empty subsets, which makes the exhaustive exploration of all subsets impossible for most libraries.

We use three libraries, referred to as L1, L2 and L3. Their parameters are summarized in table 1. L1 was selected to be small enough for allowing a complete enumeration. L2 was designed as a superset of L1: all strands from L1 can be expressed as an L2 strand. However, L2 is too large for a complete exploration and serves as a control to check that MAP-Elites can find results similar to or better than

**Table 1.** Table of simulated libraries. Each library uses different sets of domains. As such, the domain-level exploration methodology has to handle search spaces of varying dimension. L1 has a relatively small search space and can be explored exhaustively. This is not the case for L2 and L3 which are explored through MAP-Elites with a fixed budget of evaluated points.

| name | nr. domains per strand | domains size | dimension | search type | evaluation budget |
|------|------------------------|--------------|-----------|-------------|-------------------|
| L1 | 2 | a : 10, b : 10 | 16 | Exhaustive | 6 5535 |
| L2 | 4 | a : 5, b : 5 | 256 | MAP-Elites | 300 000 |
| L3 | 3 | a : 17, b : 17, c : 17, d : 15, e : 17, f : 17 | 1728 | MAP-Elites | 300 000 |

those of L1. Finally, L3 was designed to include the strands of the DNA tetrahedron [18], which ensures the existence of CRNs capable of generating complex structures.

## 2.2. Peppercorn enumerator

We use Peppercorn, an enumerator for DNA strand displacement reactions developed by Badelt *et al.* [22]. Peppercorn computes all possible domain-level interactions and pathways among a given set of starting DNA structures, with newly found structures added to the working set. Peppercorn thus iteratively computes the reaction network of all structures that can arise from the starting set.

Note that such reaction networks are very coarse-grained. Peppercorn works at the domain level, which prevents it from taking into account partial complementarity. Since structures are explicitly represented, growth is artificially bounded to keep the reaction network bounded. Additionally, structures that are deemed transient are prevented from reacting together. Finally, the algorithm forbids pseudoknotted structures, which is a limitation for the current application. However, we make the hypothesis that CRNs that would produce a large number of complex pseudoknotted structures would also produce non-pseudoknotted structures (as intermediates, for instance). In particular, those non-pseudoknotted structures would contain the same number of strands as their pseudoknotted counterparts. Those non-pseudoknotted structures will be picked up by Peppercorn, allowing us to further refine such CRNs at a later stage if necessary.

While those limitations may be mitigated, or even ignored, in the case of well-designed strand-displacement based systems, they have a direct impact on our approach. Nevertheless, Peppercorn still gives a good qualitative estimate of the complexity of a system arising from a given set of initial strands. Moreover, while other CRN enumerators could be used (e.g. [24]), Peppercorn has been specifically designed with DNA systems in mind, making it compatible with other tools used in our framework out of the box.

More details about the Peppercorn implementation and parameters can be found in the electronic supplementary material section.

## 2.3. Exploration with MAP-Elites

We want to explore the range of possible DNA structures obtained for different parameters and across the features of interest defined in the previous section. This problem is difficult because it is ill-defined and highly dimensional. It is difficult to predict that a particular initial CRN configuration leads to DNA structures of determined size. The search for large structures is also a highly deceptive process as small changes in the initial conditions potentially result in large changes in obtained structures [25–27].

All of these aspects suggest that an excessively large number of parameter values would need to be tested to identify interesting niches in the problem space leading to interesting structures. The execution of a single run of Peppercorn can also be computationally expensive (several seconds to more than 20 min, depending on the CRNs, and for our chosen Peppercorn hyperparameters). This renders simple landscape exploration methods (e.g. exhaustive search as in [20] or Monte Carlo methods) unsuitable because of the computational cost.

Furthermore, searching concurrently for several interesting solutions may be more appropriate for our purpose, rather than optimizing for a single good solution. Recent studies tackled the problem of discovering interesting niches in the problem spaces by using quality-diversity (QD) algorithms, a

family of optimization algorithms that searches for collections of solutions that are both diverse and high-performing (instead of just searching for one 'good' solution) [28,29]. Additionally, as this method favours exploration, it is especially suited for deceptive problems.

Here, we use the MAP-Elites algorithm [21] to efficiently explore the landscape of the problem space to search for the most interesting CRNs across different initial conditions with varying properties. This QD algorithm iteratively searches solutions and regroups them into a grid of elites. It is very useful as an exploration algorithm to find interesting solutions (with respect to an objective function, or 'fitness') across a number of properties of interest (named 'feature descriptors') that correspond to the axes of the grid. QD algorithms can be used to show how the interactions between these feature descriptors result in interesting solutions. As such, QD algorithms are often called 'illumination' algorithms as they reveal entire areas of the feature-space of a problem. These algorithms were previously used to efficiently explore the space of a variety of problems in the robotics [30,31], reinforcement learning [32–34] and video games [35,36] communities, but they were also used to explore problems in biology and chemistry [25,37,38].

In our domain-level exploration methodology, we use MAP-Elites to search the space of 'interesting' CRNs according to user-defined diversity properties (feature descriptors) or objective function to optimize (fitness). CRNs statistics are either computed directly from the basic properties of the CRN (e.g. number of initial strands) or from Peppercorn results.

CRN with a smaller number of strands are easier to test experimentally but lose expressivity. Conversely, large-sized CRN can describe more complex behaviours, which may be necessary for the beads to successfully self-aggregate into the target shape. As such, a trade-off in terms of topology complexity has to be considered, possibly after the optimization process. This substantiates methodologies that concurrently search for topologies of differing sizes.

For each library (L1–L3), we consider two cases to explore with MAP-Elites, each with a different objective function to optimize (fitness): either the mean structure size (MSS) or the entropy of reaction types (ERT). We make the hypothesis that these objective functions can be useful to find CRNs that can generate collections of relatively interesting DNA structures, by favouring structures that are diverse, sufficiently large to incorporate interesting patterns, or that react in a diversity of ways.

All cases consider the following feature descriptors for MAP-Elites (corresponding to axes of the grid), based on Peppercorn results:

— Complexity of the CRN ($k$ number of initial strands)
— Number of reactions, as provided by Peppercorn
— Number of structures, as provided by Peppercorn

These feature descriptors were chosen to provide a breath of possibilities relevant to designers. Having a limited number of initial strands is advantageous for *in vitro* implementation, but may limit the complexity of the CRNs. Having a high number of reactions shows that a system is subject to complex dynamics, while a low number of reactions can be preferable for cases where the designer wants well-formed set structures. The same observation can be made for the number of structures found in the system.

## 2.4. Results

We apply the domain-level exploration methods described previously to find interesting CRNs for a range of user-specified features of interest and arrange them into grids of elites (containing $3 \times 50 \times 55 = 8250$ bins). The evaluation budget (i.e. number of tested CRNs) for each library can be found in table 1. For each case, 10 optimization runs are computed and the grids of solutions found by each run are aggregated into a set of final grids. They can be found for each library in figure 3.

For each case, our methodology is able to find a collection of diverse CRNs, filling out a small portion of each grid: 295 bins (0.04%) for the L1-MSS and L1-ERT grids, 2057 (0.25%) for L2-MSS, 1812 (0.22%) for L2-ERT, 1326 (0.16%) for L3-MSS and 1333 (0.16%) for L3-ERT.

The solutions found using the two fitnesses (MSS and ERT) are different for most bins of the grids, with only a small number of intersections between the grids of MSS and those of ERT: 64 for L1 (mostly due to the small budget of evaluations), 4 for L2 and 0 for L3 (independent solutions on both sides, despite covering the same range of parameters).

The objective function landscapes span similar ranges for all libraries, however, high entropy region and high mean structure size region tend to be different. In terms of the spread of CRNs, we see a threshold at around 100 total reactions, where the maximum number of structures founds in a given

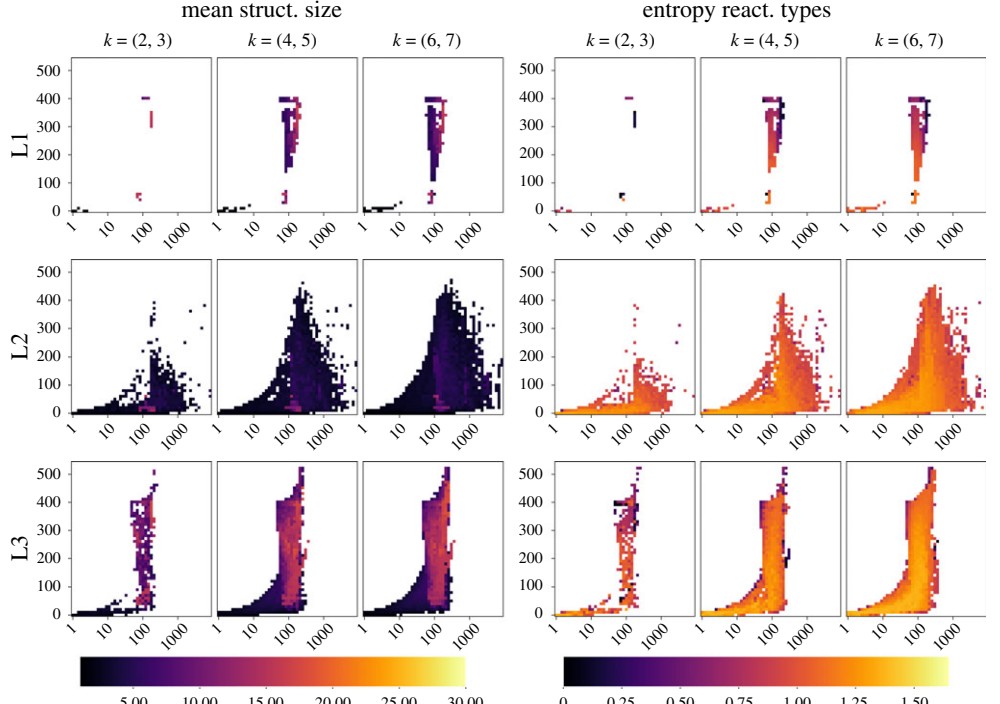

**Figure 3.** Final results of the domain-level exploration methodology. For each target library, MAP-Elites were used to find promising CRN solutions and arrange them into grids, with respect to several properties ('feature descriptors'): $k$ corresponds to the number of strands, the X and Y axes correspond respectively to the number of reactions (log10 scale) and the number of structures, as computed by the Peppercorn strand-displacement enumerator. Each bin of the $3 \times 50 \times 55$ — sized grids contains a high-performing solution according to two different metrics (fitnesses): either the mean structure size (MSS, left panel) measuring the overall tendency of generating larger structures, or the entropy of reaction types (ERT, right panel), assessing the diversity of types of reactions.

system suddenly increases. Beyond that point, the limitations of the enumeration prevent us from getting a complete view of the systems, yielding a white area in the top right of the figures.

— L1: As expected, grids are mostly empty due to the limited number of sets available. Nevertheless, we could find regions with high mean structure sizes or entropy.
— L2: The area corresponding to few structures with a high average size for roughly 100 reactions can be found in both L1 and L2. However, sets with high average size were also found in the high number of structures, high number of reactions area of L1, but not in L2, which shows that exploration is non-trivial. Results show a much better match when using the entropy of reaction types as a fitness, where all high performing regions of L1 are also found in L2.
— L3: Surprisingly, the distribution of CRNs from L3 is very close to that of L1, with CRNs found over similar feature ranges, with close objective function values.

Overall, the three libraries display similar tendencies. We can see a threshold on the number of reactions beyond which the number of different structures that can be found in a given CRN suddenly increases. While that threshold was similar for all libraries considered here, the exact value is most likely dependent on the domain design.

# 3. Sequence-level exploration

In this section, we aim to optimize the sequence used for each domain of the CRNs found in the previous section to maximize the number of well-formed structures they can generate. In this context, well-formed structures mean structures that fold according to the domain-level design. By contrast, a structure that is not well-formed would have strands hybridizing partially to incompatible domains. We describe how the number of well-formed structures can be computed, by comparing results from Peppercorn with those of another software: NUPACK [23]. We then use exploration algorithms to search for the best-performing set of sequences.

## 3.1. NUPACK

NUPACK is a suite of software for the design and analysis of secondary structures in nucleic acids. Here, we use it to generate a list of the minimum free energy (MFE) structures from a set of DNA strands. The sequence of those strands is generated from converting the domains into sequences.

To determine if a structure is well formed, we check that all hybridized nucleotides correspond to compatible domains and that they are correctly aligned. Two domains are considered hybridized if enough of their nucleotides are hybridized. The exact thresholds are given in the electronic supplementary material.

## 3.2. Matching structures

We use the previous step to rewrite the MFE structures in the kernel notation. We then match those against the structures predicted by Peppercorn, checking for possible circular permutation in the order of strands in the structure. This step gives us a number of structures that both NUPACK and Peppercorn agree on.

## 3.3. Exploration algorithms

The objective function maximized during the optimization process for a given CRN $i$ corresponds to the number of well-formed structures generated by $i$ that both NUPACK and Peppercorn agree on, for a tested set of sequences. We consider two optimization algorithms: either simple random search (i.e. new random individuals are generated and evaluated, only the best are kept) or a genetic algorithm (GA). The GA applies mutations that perform one to three random changes in the DNA bases of an individual.

More details about the sequence-level exploration can be found in electronic supplementary material along with a list of hyperparameters.

## 3.4. Results

We consider the sequence-level methodology described earlier. As this process is computationally expensive, we only apply it on a set of 12 CRNs of interest (named respectively A–L) from the final grids obtained through the domain-level exploration. These 12 CRNs are empirically selected to gather a portfolio of diverse solutions exhibiting very different properties, as a way of showcasing our exploration methodology. Alternatively, all further analyses could be rendered more exhaustive (and far more computationally expensive) by analysing all CRNs found in the final grids, and not just a sub-set of 12 individuals.

Table 2 lists the results of the sequence-level exploration process for all 12 CRNs of interest, with the scores of the best-performing set of sequences found either by random search or by the GA.

While the exploration process computes the overlap between Peppercorn and NUPACK results mostly for very small structures (NUPACK is launched with a maximum structure size of 3 to reduce computational costs), we recompute the overlap for the best-performing solutions found for larger structures (NUPACK launched with a maximum structure size of 8).

For most CRNs of interest, GA was able to find the set of sequences with the highest overlap scores (B, C, E, F, G, H, I, J, K, L) during the exploration process. However, after reevaluating these solutions to take into account larger structures size, the solutions found by GA are generally outperformed by those found by random search (A, B, C, D, E, H, I, J, K).

The best-performing set of sequences are listed in electronic supplementary material.

# 4. Network and structure analyses

We analyse the graph topology and properties of the 12 CRNs of interest selected during the domain-level exploration process from Peppercorn statistics. We are particularly interested in explaining how the initial structures can react and transform into well-formed structures.

As our interest caters to the well-formed structures of the CRNs, and how they can appear through a series of reactions from initial structures, we consider two networks for each CRN:

**Table 2.** Results of the sequence-level exploration methodology over all CRNs of interest (A–L), selected from the domain-level grids of figure 3 (MSS, mean struct size; ERT, entropy reaction types) at various coordinates (cf. electronic supplementary material, figure S1). The sets of sequences used by each CRN of interest are optimized with respect to the proportion of well-formed structures generated (corresponding to the overlap between Peppercorn and NUPACK results with max size = 3) over 10 optimization runs, either with a GA or random search. We only consider the best-found sets of sequences, and reevaluate them using NUPACK with max size = 8. Solutions with the best proportion of well-formed structures are selected for further analyses.

| | | A | B | C | D | E | F | G | H | I | J | K | L |
|---|---|---|---|---|---|---|---|---|---|---|---|---|---|
| domain-lvl. | grid | L1-ERT | L1-ERT | L1-MSS | L1-MSS | L2-ERT | L2-ERT | L2-MSS | L2-MSS | L3-ERT | L3-ERT | L3-ERT | L3-MSS |
| optimization | coordinates | 1,22,39 | 1,24,20 | 0,23,5 | 2,13,3 | 1,36,11 | 0,21,5 | 2,31,18 | 1,36,12 | 0,25,2 | 1,28,33 | 1,29,46 | 1,30,35 |
| prop. of overlaps (max struct. size 3) | Random | **70.796** | 48.052 | **57.143** | **43.966** | 22.857 | 46.154 | 18.103 | 32.558 | 69.388 | 32.624 | 61.409 | 18.056 |
| | GA | 63.333 | **65.625** | **57.143** | 42.553 | **78.571** | 46.154 | 18.103 | **78.571** | **70.213** | **68.462** | **68.169** | **25.641** |
| prop. of overlaps (max struct. size 8) | Random | **3.4206** | **0.30642** | **4.0481** | **0.025385** | **0.023117** | 0.15146 | 0.0094553 | **0.14296** | **0.29705** | **0.013003** | **0.021727** | 0.031462 |
| | GA | 0.45631 | 0.21685 | 3.2801 | 0.019575 | 0.017715 | **0.17016** | 0.0094553 | 0.11351 | 0.16165 | 0.0026326 | 0.020983 | **0.037092** |
| nr. of overlaps (max struct. size 8) | Random | 9964 | 298 | 37 | 92 | 17 | 12 | 28 | 21 | 51 | 72 | 1126 | 23 |
| | GA | 349 | 178 | 37 | 103 | 12 | 12 | 28 | 14 | 48 | 236 | 1545 | 30 |
| selected for further analyses | | random | random | random | random | random | GA | GA | random | random | random | random | GA |

— **Full network** which corresponds to the CRN with all structures (nodes) and reactions (edges), as provided by Peppercorn

— **Overlap network** which is the sub-graph containing the initial structures (source species), the well-formed structures, and all six paths between them (reactions, and intermediary structures).

Computing statistics over the full network is more computationally expensive than for the overlap network as it is composed of a higher number of nodes and edges. Conversely, results obtained from the overlap network are closer to what would happen in actual experiments, as it focuses on well-formed structures.

Our analyses will assess a number of graph properties of the CRNs (either with the full network or the overlap network), as described in the following paragraphs.

## 4.1. Robustness to removal of reactions

First, we want to assess the relative robustness of the CRNs with respect to the potential removal of reactions. Several reactions and structures of the CRN may not happen in experiments, so we want to estimate how likely the well-formed structures of the CRN are to be present and observable in actual experiments and how resilient are CRNs to the absence of these reactions and structures. It is possible to quantify this property globally, and thereby identify the likelihood that all reactions of the network will happen in actual experiments. Here, this is done by relying on three methods: (1) by checking if the network is scale-free, as scale-free networks are known to exhibit strong fault-tolerant properties to random node failure (this was shown using percolation theory in [39,40]); (2) by using connectivity metrics, like density or the distribution of degrees; (3) by computing the assortativity coefficient of the network [41], which corresponds to the propensity of nodes to connect to others with a similar degree. Density is associated with the capability of the network to transmit information efficiently because it can diffuse from one node to a large number of others. A lower density would correspond to a lower robustness to failure, especially if the missing nodes are hubs.

Second, we can assess this robustness at local (i.e. nodes or edges) levels by using network centrality measures to find the most important intermediary nodes (hubs) and edges of the network, that facilitate the transition from initial structures to well-formed structures. Here, we employ the following three commonly used centrality measures. (A) We consider the betweenness centrality of each node [42]: for a focal node, it is computed by counting the number of shortest paths passing through this node. Between two nodes of a directed graph, there exists at least one shortest path connecting them such that the number of edges of these paths is minimized. (B) We consider the edge betweenness centrality [43]: it is computed by counting the number of shortest paths passing through a focal edge. (C) We use the eigenvector centrality [44] which assesses the importance of a node with respect to the importance of its neighbours: it corresponds to the principal eigenvector of the adjacency matrix. Eigenvector and betweenness centrality are complementary: betweenness centrality provides a reliable assessment of centrality that mostly takes into account direct connections with eigenvector centrality and includes information both from direct and indirect connections of every length. In all three cases, a high centrality score is associated with more influence over the network. If a high centrality node or edge is missing during an actual experiment, it will impact the result far more than if a low centrality node or edge is missing.

## 4.2. Topological modularity

Third, we can investigate the modularity properties of the CRN, i.e. the presence or not of communities in the network. This is also related to the robustness assessed in the previous sections, but at the community level rather than at the global or more local level. This is quantified through a global clustering metric called the network average clustering coefficient [45] (i.e. the mean clustering coefficient of each node), which corresponds to the overall level of clustering of the network and its degree of modularity [45,46].

## 4.3. Topological hierarchy

Fourth, we examine the networks to check if they follow a hierarchical organization. We are especially interested in measuring the degree of flow hierarchy, typically associated with directed and self-organizing networks such as neural networks, information processing networks, etc [47]. Networks

with flow hierarchical organization have nodes that can be gathered into different levels in a way that nodes influenced by a given node (by being connected through a directed edge) are always at lower levels. The order of these levels is determined by the direction of the flows of resources throughout the network. These flows are essential to the network as they direct necessary resources for the nodes (i.e. here, chemical species) to be produced and sustained. Their presence may result in the existence of coevolving entities that may self-organize into networks with a flow hierarchy. Namely, a hierarchy emerges through the presence of nodes that strongly influence other nodes locally and globally [48,49]. Identifying if there is a flow hierarchy in the network is especially important in our case, to check if these nodes with strong influence exist, and understand how CRNs behave. Flow hierarchy has been computed in different ways in the literature. Here, we consider two complementary measures of hierarchy: global reaching centrality [49] (based on statistics on the number of nodes reachable from other nodes) and flow hierarchy [47] (based on the fraction of edges not participating in cycles).

## 4.4. Well-formed structures

The list of well-formed structures for each CRN is computed during the sequence-level exploration process and corresponds to the overlaps between structures enumerated by Peppercorn and those found by NUPACK (with max. structure size = 8). As CRNs involve reactions between several species, it can be argued that they should be represented as a directed hyper-graph. However, hyper-graphs are far harder to analyse than regular graphs: it is arguably more difficult to find simple paths and compute centrality measures for hyper-graphs [50,51], so it is also difficult to compute most other statistics. As such, we represent CRNs as directed graphs with edges corresponding to reactions between two structures, regular nodes corresponding to structures, and special nodes acting as reactions between more than two structures.

## 4.5. Results

We focus our analysis on the 12 CRNs of interest selected previously, with the best-performing sets of sequences found during the sequence-level exploration process.

The results of all analyses for each CRN of interest are shown for the full network in table 3 and for the overlap network in table 4.

There is a large variability in the numbers of nodes and edges of the 12 full networks, with E and H being the largest networks with more than 2500 nodes and more than 10 000 edges, while D is the smallest with only 142 nodes and 271 edges. This variability in size is also present in overlap networks: between 54 nodes and 118 edges for C to 980 nodes and 3166 edges for K. The proportion of overlaps varies depending on the CRN, so the sizes of overlap networks compared to full networks are not proportional, with A decreasing its number of nodes more than six times, while K only loses around 10% of its nodes.

The full network of the D, F and L CRNs exhibit higher densities compared to the other CRNs. Conversely, the overlap network of the A, C and K CRNs have a higher density than the others. This shows that these CRNs will be more robust to failure and uncertainty compared to the others. Here, we are mostly interested in finding CRNs that can reliably generate well-formed structures, so the results from the overlap network should be favoured over those from the full network when comparing network resilience to failures.

The full networks and overlap networks of all CRNs show disassortative mixing (with assortativity coefficients below 0) as high degree nodes tend to connect to low degree nodes. Disassortative networks tend to exhibit a high robustness of connectivity with respect to random node failure and a low robustness of connectivity with respect to high-degree nodes. For these networks, a relative decrease in assortativity would translate into a higher efficiency for information diffusion and an increase of the concentration of communication flows on a few edges [53,54]. These effects are particularly prevalent for CRNs with overlap networks displaying low assortative coefficients like C, E and K.

None of the CRNs of interest are scale-free (both for the full networks and for the overlap networks), as evidenced by the results of the Kolmogorov–Smirnov tests [52] (with a $p = 0$ for most cases). Note that this is not especially surprising, as scale-free networks are rare in nature [55]. Consequently, CRNs do not reap the benefits of the high robustness properties of scale-free networks to random node failures (while still retaining those of disassortative mixing).

**Table 3.** Statistical network analyses of the CRNs of interest (A–L). We characterize the networks in terms of robustness to failure (density), check if scale-free, i.e. check if the distribution of degrees follows a powerlaw with the Kolmogorov–Smirnov test [52], assortativity), modularity (global clustering) and hierarchy (global reaching centrality, flow hierarchy). All networks are disassortative and none of them are scale-free.

| name | nr. nodes | nr. edges | density | assortativity | KS powerlaw | global clustering | global reaching centrality | flow hierarchy |
|---|---|---|---|---|---|---|---|---|
| A random | 1052 | 1945 | 0.001759 | −0.08345 | $D = 0.788\ p = 0.000 \times 10^{+00}$ | $0.0 \pm 0.0$ | 0.9317 | 1.0 |
| B random | 1045 | 2268 | 0.002079 | −0.249 | $D = 0.9426\ p = 0.000 \times 10^{+00}$ | $0.0006446 \pm 0.01457$ | 0.5149 | 0.4771 |
| C random | 832 | 2302 | 0.00333 | −0.7319 | $D = 1.0\ p = 0.000 \times 10^{+00}$ | $0.0 \pm 0.0$ | 0.7031 | 1.0 |
| D random | 142 | 271 | 0.01354 | −0.1992 | $D = 0.9437\ p = 8.220 \times 10^{-178}$ | $0.0 \pm 0.0$ | 0.8346 | 0.6458 |
| E random | 2555 | 11785 | 0.001806 | −0.6667 | $D = 1.0\ p = 0.000 \times 10^{+00}$ | $0.06283 \pm 0.1086$ | 0.04733 | 0.03224 |
| F GA | 351 | 949 | 0.007725 | −0.2599 | $D = 0.9943\ p = 0.000 \times 10^{+00}$ | $0.05763 \pm 0.1219$ | 0.3118 | 0.3098 |
| G GA | 1166 | 4285 | 0.003154 | −0.4727 | $D = 0.9974\ p = 0.000 \times 10^{+00}$ | $0.06513 \pm 0.09987$ | 0.2399 | 0.07981 |
| H random | 2524 | 10526 | 0.001653 | −0.5438 | $D = 0.9996\ p = 0.000 \times 10^{+00}$ | $0.02358 \pm 0.06164$ | 0.05883 | 0.03905 |
| I random | 1135 | 2508 | 0.001949 | −0.1905 | $D = 0.9982\ p = 0.000 \times 10^{+00}$ | $0.0005952 \pm 0.009879$ | 0.1167 | 0.2185 |
| J random | 1210 | 3501 | 0.002393 | −0.2315 | $D = 0.9975\ p = 0.000 \times 10^{+00}$ | $0.003318 \pm 0.03303$ | 0.7996 | 0.5893 |
| K random | 1086 | 3754 | 0.003186 | −0.6094 | $D = 1.0\ p = 0.000 \times 10^{+00}$ | $0.008678 \pm 0.04244$ | 0.8941 | 0.4992 |
| L GA | 682 | 2944 | 0.006339 | −0.2414 | $D = 0.9941\ p = 0.000 \times 10^{+00}$ | $0.008384 \pm 0.06114$ | 0.7931 | 0.2031 |

**Table 4.** Statistical analyses of the CRNs of interest (A–L) over the 'overlap networks', i.e. the sub-graph containing the source species, all well-formed species, and the six paths between them, including intermediary species. We characterize the network in terms of robustness to failure (density), check if scale-free, i.e. check if the distribution of degrees follows a powerlaw with the Kolmogorov–Smirnov test [52], assortativity), modularity (global clustering) and hierarchy (global reaching centrality, flow hierarchy). All networks are disassortative and none of them are scale-free.

| name | nr. nodes | nr. edges | density | assortativity | KS powerlaw | global clustering | global reaching centrality | flow hierarchy |
|---|---|---|---|---|---|---|---|---|
| A random | 156 | 290 | 0.01199 | −0.1677 | $D = 0.8718\ p = 1.946 \times 10^{-139}$ | $0.0 \pm 0.0$ | 0.8467 | 1.0 |
| B random | 713 | 1501 | 0.002957 | −0.2053 | $D = 0.9509\ p = 0.000 \times 10^{+00}$ | $0.0 \pm 0.0$ | 0.3741 | 0.3504 |
| C random | 54 | 118 | 0.04123 | −0.606 | $D = 1.0\ p = 0.000 \times 10^{+00}$ | $0.0 \pm 0.0$ | 0.6849 | 1.0 |
| D random | 142 | 271 | 0.01354 | −0.1992 | $D = 0.9437\ p = 8.220 \times 10^{-178}$ | $0.0 \pm 0.0$ | 0.8346 | 0.6458 |
| E random | 689 | 2393 | 0.005048 | −0.458 | $D = 0.9971\ p = 0.000 \times 10^{+00}$ | $0.05465 \pm 0.1247$ | 0.08563 | 0.03385 |
| F GA | 243 | 679 | 0.01155 | −0.2456 | $D = 0.9918\ p = 0.000 \times 10^{+00}$ | $0.07885 \pm 0.1387$ | 0.08422 | 0.1502 |
| G GA | 331 | 898 | 0.008221 | −0.2696 | $D = 0.9849\ p = 0.000 \times 10^{+00}$ | $0.05396 \pm 0.09722$ | 0.4839 | 0.3408 |
| H random | 446 | 1376 | 0.006933 | −0.2498 | $D = 0.9955\ p = 0.000 \times 10^{+00}$ | $0.05518 \pm 0.1156$ | 0.2058 | 0.1352 |
| I random | 681 | 1379 | 0.002978 | −0.226 | $D = 1.0\ p = 0.000 \times 10^{+00}$ | $0.0002466 \pm 0.006382$ | 0.1976 | 0.2487 |
| J random | 300 | 681 | 0.007592 | −0.1903 | $D = 0.9867\ p = 0.000 \times 10^{+00}$ | $0.0 \pm 0.0$ | 0.7511 | 0.5007 |
| K random | 980 | 3166 | 0.0033 | −0.5987 | $D = 1.0\ p = 0.000 \times 10^{+00}$ | $0.009124 \pm 0.04432$ | 0.8963 | 0.5401 |
| L GA | 155 | 369 | 0.01546 | −0.3084 | $D = 0.9742\ p = 1.315 \times 10^{-246}$ | $0.01039 \pm 0.08964$ | 0.6363 | 0.5176 |

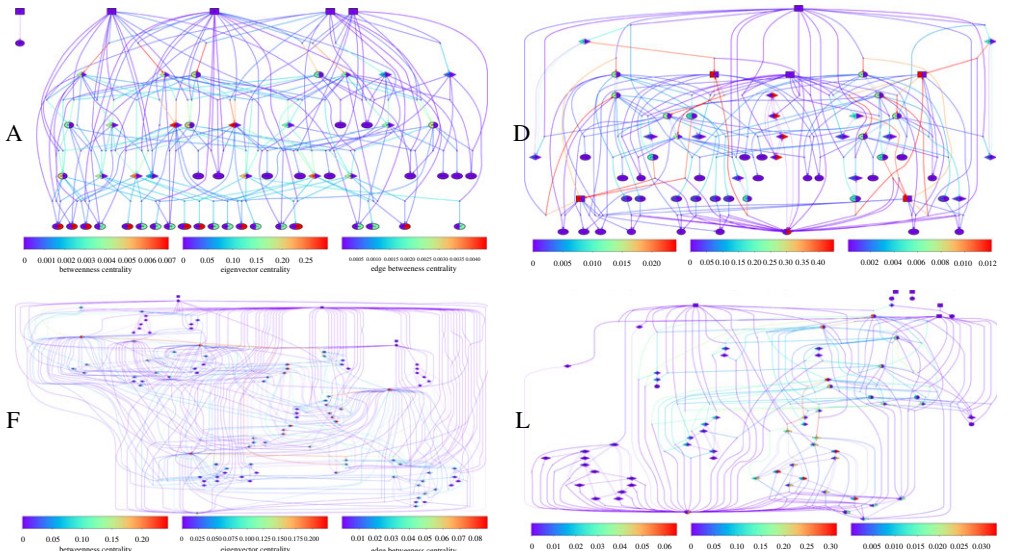

**Figure 4.** Simplified networks of the A, D, F, L CRNs of interest. We only show the 'overlap networks' containing the source species (rectangular nodes), the 50 first well-formed species (ellipses), and the six paths between them, including intermediary species (diamond). The reactions are expressed as point nodes. Scores of betweenness and eigenvector centrality are represented as coloured parts of each node, respectively on the left and right sides. Edges coloration quantify edge betweenness centrality.

Overlap networks of the A, D, F and L CRNs are presented in figure 4 (other CRNs are shown in electronic supplementary material). As these networks tend to be very large, we only show simplified versions with only the first 50 well-formed species. These networks describe how well-formed structures can emerge through hierarchical self-assembly from the source structures. Each network has coloured nodes and edges to indicate their scores of betweenness centrality, eigenvector centrality, and edge betweenness centrality. Note that, as all networks are disassortative, the most vulnerable nodes and edges are those with the highest degree; other measures of centrality are still important to assess which nodes and edges are the most important for the flow of information in the networks, but they are less relevant to identify which nodes and edges failure would impact most the integrity of the overall networks.

The networks for all CRNs display very little (to none) global clustering. This indicates that nodes do not tend to create groups with relatively high local density, and suggests that the networks do not exhibit modular topologies.

All networks display varying levels of hierarchy. When comparing two networks, higher scores correspond to a relatively higher level of hierarchy. Here, global reaching centrality and flow hierarchy scores both give very similar scores for each CRN, both for full networks and for overlap networks. This is not surprising as both rely on notions of flow hierarchy. These scores suggest that the level of hierarchy is very dependent on the library: most CRNs of the L1 library exhibit scores are above 0.60 (except B), all scores of L2 CRNs are below 0.50, and most scores of the L3 CRNs are above 0.50 and below 0.90 (except I). High levels of flow hierarchy would also suggest that several CRNs are organized into 'layered' sets of nodes with a common underlying direction.

Additional statistics (including the distribution of degree) are available in electronic supplementary material.

Figure 5 shows two-dimensional representations with helicity of several instances of the largest structures generated by each CRN, rendered by the NUPACK software. Note that only well-formed structures that both NUPACK and Peppercorn agree on were selected. This representation was chosen as a proxy for three-dimensional structures that remains tractable in terms of computational cost.

Sets from the L1 library (A, B, C and D), having only two domains per strand, mostly generate linear polymer structures. When structures are long enough to be flexible, NUPACK predicts that they may fold onto themselves (B, D). Structures generated with the objective function set to maximize the entropy of reaction diversity (C, D) have defects, as those provide toeholds and increase interactions between structures.

Sets from the L2 library (E, F, G, H) have a similar tendency to those of the L1 library. However, the larger number of domains per strand offers more flexibility, which is reflected in the diversity of the structures found.

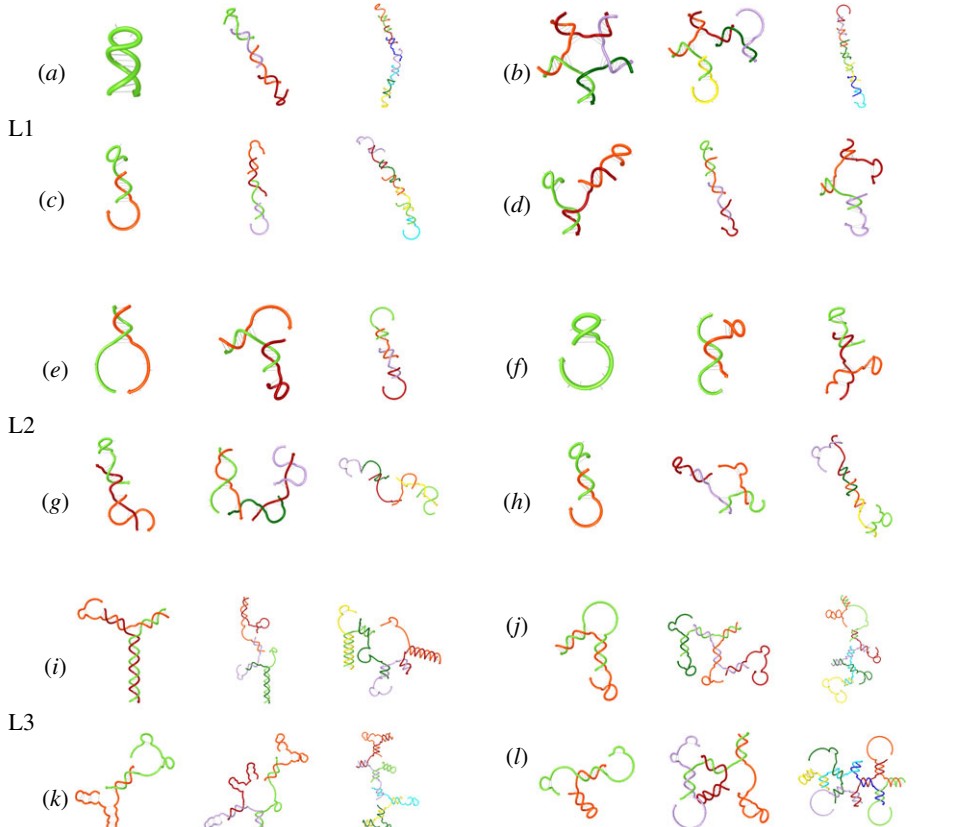

**Figure 5.** Instances of the most representative structure families generated by all CRNs of interest depicted with helicity by the NUPACK software.

Finally, sets from the L3 library (I, J, K, L) generated the most diversity in terms of structures. In particular, those structures tend to display complex branching patterns that are not found in L1 or L2. Based on those results, a certain number of domain types might be required to reach a certain level of complexity.

# 5. Discussion and conclusion

In this paper, we proposed a framework capable of efficiently exploring the categories of structures that can be created by folding from a given library of DNA strands. The information gathered through this process can then characterize the condition for the emergence of such structures, providing relevant insight to the designer. The workflow can also be used as a design tool, by using an objective function to search for CRNs generating target structures.

Our framework works at three levels: domain level, network level and network analysis. All levels, respectively, provide relevant clues as to the emergence of various categories of structures and can be used at different stages of the design process.

At the domain level, our methodology efficiently explores the space of CRNs (with abstract strands) created from a given library through the MAP-Elites algorithm. The resulting information identifies the categories of CRNs that can be created from this library and arrange each category into a grid according to a number of user-defined properties of interest ('features descriptors', such as the mean size of generated structures, or statistics on the number of types of reactions).

At the sequence level, our framework adapts the results of the domain-level exploration towards actual experimental design by calibrating (either through evolutionary search or random search) the abstract-stranded CRNs to find the best-performing set of sequences that generates the highest number of well-formed structures (i.e. structures that fold according to the domain-level design).

Finally, at the network analysis level, these CRNs are analysed to identify which intermediary structures are the most important to the generation of well-formed structures and quantify the CRNs' general robustness to reaction failure. This assesses how likely the well-formed structures can be observed in actual experiments.

We compared three different libraries of DNA strands, with different characteristics. The simplest library, L1, had a limited number of strands, which allowed us to perform an exhaustive search. The second library, L2, was a superset of L1, but too large for an exhaustive search. Exploration of L2 showed that our quality diversity algorithm was able to find a diversity of behaviours, including most of those found in L1. Finally, a third library, L3, had a larger number of domains. Structures generated from the CRNs made from L3 had more diversity than those of L1 or L2, highlighting the impact of the number of orthogonal domains available on diversity. Indeed, increasing that number much higher would allow the generation of arbitrary structures, in a fashion similar to DNA bricks [7,8], at the cost that the number of dimensions in the problem would make exploration difficult. Further investigation of that phenomenon could provide more precise conditions for the transition to a state where a large number of different structures can emerge from a limited set of strands. Additionally, a better understanding of such transitions will give designers a choice between making a small set of structures or going for open-ended evolution [56]. In a similar fashion to DNA bricks, we limited our exploration to libraries of strands with an equal number of domains. However, our framework supports libraries of strands of diverse lengths. In particular, designs such as the buckyball from He *et al.* are promising examples of the structures that can be created with such libraries. DNA origami are another example of such structures, where a very long scaffold strand is combined with short (2–4 domains) staple strands. Extending the approach presented in this paper, one could search for sets of staples that can fold into multiple structures, similar to the concept of Niekamp *et al.* [12]. Such exploration does, however, come at the cost of an exponential increase in the size of the search space and must thus be coupled with further methods to constrain the exploration.

All analyses of structures were performed on either enumeration of reactions based on Peppercorn or on theoretically favourable structures based on NUPACK. As such, the results are artificially limited by the absence of pseudoknots in Peppercorn and the lack of analysis of the kinetics of the system. Simulation of the dynamics of the CRNs generated could provide insights on the structures that might actually emerge in an experimental setting. Additionally, the most promising sets of strands, or those that are expected to have a large number of pseudoknots, could be simulated with a more explicit physical model, such as OxDNA [57]. OxDNA simulates DNA strands as three-dimensional objects subjected to a variety of forces and thus do not have any limitation with respect to the presence of pseudoknots. The downside of that approach is that the simulation of systems is stochastic and would thus neither generate an actual CRN nor prove the absence of interesting structures in a given system. Simulations are extremely costly and can take up to 5 days for 300 ns of simulated time on a dedicated computing server for large structures [58], preventing the application of that simulator at an earlier stage of the exploration. Alternatively, a different enumerator might be plugged in the framework instead of Peppercorn, or as an additional refinement stage for the CRN exploration.

Another possible perspective is to use an automated experimental set-up [59–62] to explore the impact of the concentrations of the different strands in a given system. Based on the results of He *et al.* [13], we expect to find systems with a large variety of potential behaviours selected through initial conditions.

Our graph analysis uncovers interesting network properties of CRNs, including their disassortative topologies, where a large number of reactions pass through a small number of high-degree nodes. CRNs should be robust to random failure (i.e. if some random reactions enumerated by Peppercorn do not actually happen during experiments), but be especially vulnerable to the failure of the most important nodes, including high-degree nodes and edges. CRNs have a low global clustering, suggesting that they are not modular. CRNs exhibit varying degrees of hierarchy, with results suggesting that hierarchy is related to the library used.

Our graph analysis could be improved by not only taking into account the topology of the networks, but also the rates of the reactions (weights of the edges). These values could be retrieved by recent versions of the Peppercorn software that include heuristics to approximate them. Here, we represent CRNs as directed graphs, while a directed hyper-graph representation could arguably be more adapted to describe reactions involving more than two species. This simplifying choice was motivated by the complexity of hyper-graph algorithms. Alternatively, our approach could be improved by computing hyper-graph statistics.

Both the domain-level and sequence-level exploration methodologies are computationally expensive because they need to execute a large number of instances of Peppercorn and NUPACK. Reducing the computational costs of each instance would allow the methodology to be executed on larger budgets of evaluations to explore further and find more interesting solutions. In the domain-level exploration case, it would also be possible to render the search process more sample-efficient by using recent

improvements to the MAP-Elites algorithm, possibly relying on surrogate models [63] or feature extraction methods [64–66], or on more complex search schemes [35,67]. The search process may also benefit from the use of 'intelligent mutation' operators with a more complex understanding of the network topology of the CRNs (here, mutations are merely random bit-flips over a Boolean vector).

Finally, the proposed approach provides a way to implement compotypes [68]. A compotype is a theoretical quasi-stable agglomeration of molecular species that can grow and divide, a mechanism that allows them to multiply. With our approach, we can look for families of structures that interact with each other and single strands present in the system in a way similar to compotypes. By further functionalizing those structures [69] (e.g. providing them sensing or actuation), one could define populations of molecular robots with the ability to multiply. Such capability would provide them with a better robustness to defects, as the number of robots could dynamically increase when needed, or better adaptability, as external signal could trigger a shift in the dominant structures in the system, thus 'recycling' one type of robot into another.

Overall, we showed that our methodology is able to gather relevant insights both at the domain level, sequence level and network level to characterize the condition for the emergence of a diverse collection of structure families. It can thus be used as a framework to drive the nanostructure design process.

Data accessibility. All scripts used in this paper are available at https://bitbucket.org/leo-cazenille/kakenhievolvedna.git and have been archived within the Zenodo repository, DOI: 10.5281/zenodo.5462259 [70]. The MAP-Elites algorithm [21] used in this paper was implemented as part of the QDpy library [71], available freely at https://gitlab.com/leo.cazenille/qdpy.

Authors' contributions. L.C. and N.A.K. designed the workflow, wrote the code and performed data analysis. L.C. computed the simulations. A.B. and N.A.K. designed the DNA libraries used for exploration. All authors wrote and proof-read the manuscript.

Competing interests. We declare we have no competing interests.

Funding. This work was supported by JSPS KAKENHI grant no. JP17K00399 and by Grant-in-Aid for JSPS Fellows JP19F19722.

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
