## [Peer Review File · Royal Society Open Science]

Review History

RSOS-210848.R0 (Original submission)

Review form: Reviewer 1

Is the manuscript scientifically sound in its present form?

Yes

Are the interpretations and conclusions justified by the results?

Yes

Is the language acceptable?

Yes

Do you have any ethical concerns with this paper?

No

Have you any concerns about statistical analyses in this paper?

No

Recommendation?

Accept with minor revision (please list in comments)

Comments to the Author(s)

This manuscript described the use of automated tools for state-space exploration to characterize DNA-based chemical reaction networks for the assembly of DNA-based nanostructures. The general idea, though I do not think that this is stated quite so explicitly in the paper, is that these CRNs model the set of assembly (and perhaps disassembly) pathways possible for a nanostructure constructed from strands drawn from each of the strand sets under study. This is a somewhat unusual approach to the problem of DNA nanostructure assembly, which usually proceeds by settling on a particular target structure ahead of time and designing strands that program a pathway to assemble into that structure (and, hopefully, no others). Here, however, the authors take the alternative approach of trying to characterize the assembly landscape of various "libraries" of DNA strands.

Based on reading the authors response to reviewers document supplied with the manuscript, it seems that they have done a fairly good job of responding to the issues raised in the previous set of reviews. The paper introduction is quite good, in particular. One particular issue that could be addressed more directly is the restriction of the Peppercorn reaction enumerator to non-pseudoknotted structures, which would seem to rule out a lot of possible structures of interest. The authors do mention this in passing, but it would be good to think about if there are classes of structures of practical interest that fall within the representable set of this system in a revised version.

In any case, while the practicality of this approach for general-purpose molecular design may be debated, this manuscript does outline a quite thorough study of this particular dataset of strand interactions that is likely to be of interest to molecular designers. Therefore it is appropriate for publication in Royal Society Open Science.

Review form: Reviewer 2

Is the manuscript scientifically sound in its present form?

Yes

Are the interpretations and conclusions justified by the results?

Yes

Is the language acceptable?

Yes

Do you have any ethical concerns with this paper?

No

Have you any concerns about statistical analyses in this paper?

No

Recommendation?

Accept as is

Comments to the Author(s)

The authors have answered my questions and revised/updated the manuscript accordingly. Thus, I consider the current form of the manuscript is suitable for publication.

Decision letter (RSOS-210848.R0)

Dear Dr Aubert-Kato

On behalf of the Editors, we are pleased to inform you that your Manuscript RSOS-210848 "Automated Exploration of DNA-based Structure Self-Assembly Networks" has been accepted for publication in Royal Society Open Science subject to minor revision in accordance with the referees' reports. Please find the referees' comments along with any feedback from the Editors below my signature.

Please submit your revised manuscript and required files (see below) no later than 7 days from today's (ie 09-Jun-2021) date. Note: the ScholarOne system will 'lock' if submission of the revision is attempted 7 or more days after the deadline. If you do not think you will be able to meet this deadline please contact the editorial office immediately.

on behalf of Professor Ion Petre (Associate Editor) and Marta Kwiatkowska (Subject Editor)
openscience@royalsociety.org

Reviewer comments to Author:

Reviewer: 1

Comments to the Author(s)

This manuscript described the use of automated tools for state-space exploration to characterize DNA-based chemical reaction networks for the assembly of DNA-based nanostructures. The general idea, though I do not think that this is stated quite so explicitly in the paper, is that these CRNs model the set of assembly (and perhaps disassembly) pathways possible for a nanostructure constructed from strands drawn from each of the strand sets under study. This is a somewhat unusual approach to the problem of DNA nanostructure assembly, which usually proceeds by settling on a particular target structure ahead of time and designing strands that

program a pathway to assemble into that structure (and, hopefully, no others). Here, however, the authors take the alternative approach of trying to characterize the assembly landscape of various "libraries" of DNA strands.

Based on reading the authors response to reviewers document supplied with the manuscript, it seems that they have done a fairly good job of responding to the issues raised in the previous set of reviews. The paper introduction is quite good, in particular. One particular issue that could be addressed more directly is the restriction of the Peppercorn reaction enumerator to non-pseudoknotted structures, which would seem to rule out a lot of possible structures of interest. The authors do mention this in passing, but it would be good to think about if there are classes of structures of practical interest that fall within the representable set of this system in a revised version.

In any case, while the practicality of this approach for general-purpose molecular design may be debated, this manuscript does outline a quite thorough study of this particular dataset of strand interactions that is likely to be of interest to molecular designers. Therefore it is appropriate for publication in Royal Society Open Science.

Reviewer: 2

Comments to the Author(s)

The authors have answered my questions and revised/updated the manuscript accordingly. Thus, I consider the current form of the manuscript is suitable for publication.

===PREPARING YOUR MANUSCRIPT===

Your revised paper should include the changes requested by the referees and Editors of your manuscript. You should provide two versions of this manuscript and both versions must be provided in an editable format:
one version identifying all the changes that have been made (for instance, in coloured highlight, in bold text, or tracked changes);
a 'clean' version of the new manuscript that incorporates the changes made, but does not highlight them. This version will be used for typesetting.

===PREPARING YOUR REVISION IN SCHOLARONE===

Author's Response to Decision Letter for (RSOS-210848.R0)

See Appendix A.

Decision letter (RSOS-210848.R1)

Dear Dr Aubert-Kato,

I am pleased to inform you that your manuscript entitled "Automated Exploration of DNA-based Structure Self-Assembly Networks" is now accepted for publication in Royal Society Open Science.

on behalf of Professor Ion Petre (Associate Editor) and Marta Kwiatkowska (Subject Editor)
openscience@royalsociety.org

Appendix A

Reviewer 1:

>One particular issue that could be addressed more directly is the restriction of the Peppercorn reaction enumerator to non-pseudoknotted structures, which would seem to rule out a lot of possible structures of interest. The authors do mention this in passing, but it would be good to think about if there are classes of structures of practical interest that fall within the representable set of this system in a revised version.

In practice, we only need Peppercorn to find enough structures for a given set of strands for the set to be selected for the next step of the workflow. While we cannot distinguish between sets that produce only non-pseudoknotted structures and those that produce many pseudoknotted structures at this stage, further refinement steps, including the use of NUPACK, are expected to fill that gap. We added the following in the domain exploration section:

"However, we make the hypothesis that CRNs that would produce a large number of complex pseudoknotted structures would also produce non-pseudoknotted structures (as intermediates, for instance). In particular, those non-pseudoknotted structures would contain the same number of strands as their pseudoknotted counterparts. Those non-pseudoknotted structures will be picked up by Peppercorn, allowing us to further refine such CRNs at a later stage if necessary."

We also mentioned that the current workflow can be adapted to work with an enumerator that produce pseudoknotted structures, if one becomes available:

"Alternatively, a different enumerator might be plugged in the framework instead of Peppercorn, or as an additional refinement stage for the CRN exploration."